# Kalman Normalization: Normalizing Internal Representations Across Network Layers

**Guangrun Wang**
Sun Yat-sen University
wanggrun@mail2.sysu.edu.cn

**Jiefeng Peng**
Sun Yat-sen University
jiefengpeng@gmail.com

**Ping Luo**
The Chinese University of Hong Kong
pluo.lhi@gmail.com

**Xinjiang Wang**
SenseTime Group Ltd.

**Liang Lin** *
Sun Yat-sen University
linliang@ieee.org

## Abstract

As an indispensable component, Batch Normalization (BN) has successfully improved the training of deep neural networks (DNNs) with mini-batches, by normalizing the distribution of the internal representation for each hidden layer. However, the effectiveness of BN would diminish with the scenario of micro-batch (*e.g.* less than 4 samples in a mini-batch), since the estimated statistics in a mini-batch are not reliable with insufficient samples. This limits BN's room in training larger models on segmentation, detection, and video-related problems, which require small batches constrained by memory consumption. In this paper, we present a novel normalization method, called Kalman Normalization (KN), for improving and accelerating the training of DNNs, particularly under the context of micro-batches. Specifically, unlike the existing solutions treating each hidden layer as an isolated system, KN treats all the layers in a network as a whole system, and estimates the statistics of a certain layer by considering the distributions of all its preceding layers, mimicking the merits of Kalman Filtering. On ResNet50 trained in ImageNet, KN has 3.4% lower error than its BN counterpart when using a batch size of 4; Even when using typical batch sizes, KN still maintains an advantage over BN while other BN variants suffer a performance degradation. Moreover, KN can be naturally generalized to many existing normalization variants to obtain gains, *e.g.*equipping Group Normalization [34] with Group Kalman Normalization (GKN). KN can outperform BN and its variants for large scale object detection and segmentation task in COCO 2017.

## 1 Introduction

Batch Normalization (BN) [13] has recently become a standard and crucial component for improving the training of deep neural networks (DNNs), which is successfully employed to harness several state-of-the-art architectures[8, 27]. In the training and inference of DNNs, BN normalizes the internal representations of each hidden layer by subtracting the mean and dividing the standard deviation, as illustrated in Fig. 1 (a). As pointed out in [13], BN enables using larger learning rate in training, leading to faster convergence.

Although the significance of BN has been demonstrated in many previous works, its drawback cannot be neglected, *i.e.*its effectiveness diminishing when small mini-batch is presented in training. Consider a DNN consisting of a number of layers from bottom to top. In the traditional BN, the

normalization step seeks to eliminate the change in the distributions of its internal layers, by reducing their internal covariant shifts. Prior to normalizing the distribution of a layer, BN first estimates its

statistics, including the means and variances. However, it is impractically expected that the statistics of the internal layers can be pre-estimated on the training set, as the representations of the internal layers keep changing after the network parameters have been updated in each training step. Hence, BN handles this issue by the following schemes. i) During the model training, it approximates the population statistics by using the batch sample statistics in a mini-batch. ii) It retains the moving average statistics in each training iteration, and employs them during the inference.

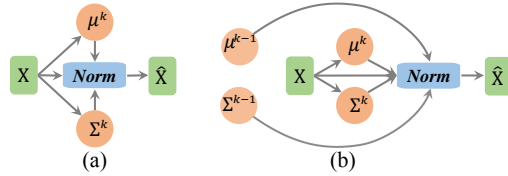

(a)                    (b)

Figure 1: (a) illustrates the distribution estimation in the conventional BN, where the mini-batch mean $\mu^k$ and variance $\Sigma^k$, are estimated based on the currently observed mini-batch at the $k$-th layer. X and $\widehat{X}$ denote the internal representation before and after normalization. In (b), the proposed KN provides more accurate distribution estimation of the $k$-th layer, by aggregating the statistics of the preceding $(k\text{-}1)$-th layer.

However, BN has a limitation, which is limited by the memory capacity of computing platforms (*e.g.*GPUs), especially when the network size and image size are large. In this case, the mini-batch size is not sufficient to approximate the statistics, making them had bias and noise. And the errors would be amplified when the network becomes deeper, degenerating the quality of the trained model. Negative effects exist also in the inference, where the normalization is applied for each testing sample. Furthermore, in the BN mechanism, the distribution of a certain layer could vary along with the training iteration, which limits the stability of the convergence of the model.

The demanding on batch size limits the performance of many computer vision task, such as detection [7, 9], segmentation [3], video recognition [28], and other high-level systems built upon them [32, 31]. For instance, limited by the heavy burden of model and the high resolution of images, the Mask RCNN frameworks [9] can only allow an extremely micro batch (*e.g.*1 or 2), which disable the function of BN as discussed above. Compromisingly, a common way is to 'freeze' the BN, in which BN degrades into a linear layer because the statistics it uses are fixed as constants.

In this paper, we present a new normalization method, called Kalman Normalization (KN), for improving and accelerating training of DNNs particularly under the context of micro-batches. KN advances the existing solutions by achieving more accurate estimation of the statistics (means and variances) of the internal representations in DNNs. Unlike BN where the statistics were estimated by only measuring the mini-batches within a certain layer, *i.e.*they considered each layer in the network as an isolated sub-system, KN shows that the estimated statistics have strong correlations among the sequential layers. And the estimations can be more accurate by jointly considering its preceding layers in the network, as illustrated in Fig. 1 (b). By analogy, the proposed estimation method shares merits compared to the Kalman filtering process [14]. KN performs two steps in an iterative way. In the first step, KN estimates the statistics of the current layer conditioned on the estimations of the previous layer. In the second step, these estimations are combined with the observed batch sample means and variances calculated within a mini-batch.

This paper makes the following contributions. 1) We propose an intuitive yet effective normalization method, offering a promise of improving and accelerating the neural network training. 2) The proposed method enables training networks with mini-batches of very small sizes (*e.g.* less than 4 examples), and the resulting models perform substantially better than those using the existing BN methods. This specifically makes our method advantageous in several memory-consuming problems such as large scale object detection and segmentation task in COCO 2017. 3) On ImageNet classification task, the experiments show that the recent advanced networks can be strengthened by our method, and the trained models improve the leading results by using less than 60% training steps. And the computational complexity of KN increases only $0.015\times$ compared to that of BN, leading to a marginal additional computation.

## 2   Related Work

**Whitening.** Decorrelating and whitening the input data [16] has been demonstrated to speed up training of DNNs. Some following methods [33, 22, 21] were proposed to whiten activations by using sampled training data or performing whitening every thousands iterations to reduce computation. Nevertheless, these operations would lead to model blowing up according to [13], because of

instability of training. Recently, the Whitened Neural Network [5] and its generalizations [18, 17, 11] presented practical implementations to whiten the internal representation of each hidden layer, and drew the connections between whitened networks and natural gradient descent. Although these approaches had theoretical guarantee and achieved promising results by reducing the computational complexity of the Singular Value Decomposition (SVD) in whitening, their computational costs are still not neglectable, especially when training a DNN with plenty of convolutional layers on a large-scale dataset (*e.g.*ImageNet), as many recent advanced deep architectures did.

**Standardization.** To address the above issues, instead of whitening, Ioffe *et al*.[13] proposed to normalize the neurons of each hidden layer independently, where the batch normalization (BN) is calculated by using mini-batch statistics. The extension [4] adapted BN to recurrent neural networks by using a re-parameterization of LSTM. In spite of their successes, the heavy dependence of the activations in the entire batch causes some drawbacks to these methods. For example, when the mini-batch size is small, the batch statistics are unreliable. Hence, several works [25, 2, 1, 26, 10, 34] have been proposed to alleviate the mini-batch dependence. Normalization propagation [1] attempted to normalize the propagation of the network by using a careful analysis of the nonlinearities, such as the rectified linear units. Layer normalization [2], Instance Normalization [29], and Group Normalization (GN) [34] standardized the hidden layer activations, which are invariant to feature shifting and scaling of per training sample. Fixed normalization [26] provided an alternative solution, which employed a separate and fixed mini-batch to compute the normalization parameters. However, all of these methods estimated the statistics of the hidden layers separately, whereas KN treats the entire network as a whole to achieve better estimations. Moreover, KN can be naturally applied to many existing normalization variants to obtain gains, *e.g.*equipping Group Normalization (GN) with Group Kalman Normalization (GKN)

## 3 The Proposed Approach

**Overview.** Here we introduce some necessary notations that will be used throughout this paper. Let $x^k$ be the feature vector of a hidden neuron in the $k$-th hidden layer of a DNN, such as a pixel in the hidden convolutional layer of a CNN. BN normalizes the values of $x^k$ by using a mini-batch of $m$ samples, $B = \{x_1^k, x_2^k, ..., x_m^k\}$. The mean and covariance of $x^k$ are approximated by

$$\bar{x}^k \leftarrow \frac{1}{m}\sum_{i=1}^{m}x_i^k, \quad S^k \leftarrow \frac{1}{m}\sum_{i=1}^{m}(x_i^k - \bar{x}^k)(x_i^k - \bar{x}^k)^T. \tag{1}$$

They are adopted to normalize $x^k$. We have $\hat{x}^k \leftarrow \frac{x_i^k - \bar{x}^k}{\sqrt{\text{diag}(S^k)}}$, where $\text{diag}(\cdot)$ denotes the diagonal entries of a matrix, *i.e.*the variances of $x^k$. Then, the normalized representation is scaled and shifted to preserve the modeling capacity of the network, $y^k \leftarrow \gamma\hat{x}^k + \beta$, where $\gamma$ and $\beta$ are parameters that are opmizted in training. However, a mini-batch with moderately large size is required to estimate the statistics in BN. It is compelling to explore better estimations of the distribution in a DNN to accelerate training.

### 3.1 DNN as Kalman Filtering Process

Assume that the true values of the hidden neurons in the $k$-th layer can be represented by the variable $x^k$, which is approximated by using the values in the previous layer $x^{k-1}$. We have

$$x^k = \mathbf{A}^k x^{k-1} + u^k, \tag{2}$$

where $\mathbf{A}^k$ is a state transition matrix (e.g. convolutional filters) that transforms the states (features) in the previous layer to the current layer. And $u^k$ is a bias following a Gaussian distribution. As the above true values of $x^k$ exist yet not directly accessible, they can be measured by the observation $z^k$ with a bias term $v^k$,

$$z^k = x^k + v^k, \tag{3}$$

where $z^k$ indicates the observed values of the features in a mini-batch. Then, the estimation of true value of the $k$-th layer's hidden neurons $\hat{x}^{k|k}$ and their variances $\hat{\Sigma}^{k|k}$ can be easily obtained by a standard Kalman filtering process:

$$\begin{cases} \hat{x}^{k|k-1} = \mathbf{A}^k \hat{x}^{k-1|k-1}, \\ \hat{\Sigma}^{k|k-1} = \mathbf{A}^k \hat{\Sigma}^{k-1|k-1}(\mathbf{A}^k)^{\mathrm{T}} + R, \\ \hat{x}^{k|k} = f(q^k, \hat{x}^{k|k-1}, z^k), \\ \hat{\Sigma}^{k|k} = g(q^k, \hat{\Sigma}^{k|k-1}, S^k), \end{cases} \tag{4}$$

where $\hat{x}^{k|k-1}$ and $\hat{\Sigma}^{k|k-1}$ are the estimation of true value and the variances of the $k$-th layer conditioned on the previous layer, respectively. $f(\cdot)$ and $g(\cdot)$ are two linear combination functions in the original Kalman filtering process. $R$ is the covariance matrix of the bias $u^k$ in Eqn.(2). $S^k$ is the observed covariance matrix of the mini-batch in the $k$-th layer. $q^k$ is the gain value.

## 3.2 Kalman Normalization

Eqn. 4 is a Kalman filtering process, in which the true value of the $k$-th layer's hidden neurons $\hat{x}^{k|k}$ and their variances $\hat{\Sigma}^{k|k}$ are estimated. But in a BN problem the desired quantity to estimate includes not just the variances, but also the means $\hat{\mu}^{k|k}$. Fortunately, the means can be easily obtained due to the Kalman filter property. Specifically, we compute expectation on both sides of Eqn.(2) and 3, $i.e. \mathbb{E}[x^k] = \mathbb{E}[\mathbf{A}^k x^{k-1} + u^k]$ and $\mathbb{E}[z^k] = \mathbb{E}[x^k + v^k]$, and have

$$\hat{\mu}^{k|k-1} = \mathbf{A}^k \hat{\mu}^{k-1|k-1}, \quad \mathbb{E}[z^k] = \overline{x}^k \tag{5}$$

where $\hat{\mu}^{k-1|k-1}$ denotes the estimation of mean in the $(k$-1$)$-th layer, and $\hat{\mu}^{k|k-1}$ is the estimation of mean in the $k$-th layer conditioned on the previous layer. We call $\hat{\mu}^{k|k-1}$ an intermediate estimation of the layer $k$, because it is then combined with the mean of observed values to achieve the final estimation. As shown in Eqn.(6) below, the estimation in the current layer $\hat{\mu}^{k|k}$ is computed by combining the intermediate estimation with a bias term, which represents the error between the mean of the observed values $\mathbb{E}[z^k]$ and $\hat{\mu}^{k|k-1}$. Here $\mathbb{E}[z^k]$ indicates the mean of the observed values and we have $\mathbb{E}[z^k] = \overline{x}^k$ in Eqn. 5. And $q^k$ is a gain value indicating how much we reply on this bias.

$$\hat{\mu}^{k|k} = \hat{\mu}^{k|k-1} + q^k(\overline{x}^k - \hat{\mu}^{k|k-1}). \tag{6}$$

Similarly, the estimations of the covariances can be achieved by calculating $\hat{\Sigma}^{k|k-1} = \mathrm{Cov}(x^k - \hat{\mu}^{k|k-1})$ and $\hat{\Sigma}^{k|k} = \mathrm{Cov}(x^k - \hat{\mu}^{k|k})$, where $\mathrm{Cov}(\cdot)$ represents the definition of the covariance matrix. By introducing $p^k = 1 - q^k$, and combining the above definitions with Eqn.(5) and (6), we have the following update rules to estimate the statistics as shown in Eqn.(7).

$$\begin{cases} \hat{\mu}^{k|k-1} = \mathbf{A}^k \hat{\mu}^{k-1|k-1}, \\ \hat{\mu}^{k|k} = p^k \hat{\mu}^{k|k-1} + q^k \overline{x}^k, \\ \hat{\Sigma}^{k|k-1} = \mathbf{A}^k \hat{\Sigma}^{k-1|k-1} (\mathbf{A}^k)^{\mathrm{T}} + R, \\ \hat{\Sigma}^{k|k} = p^k \hat{\Sigma}^{k|k-1} + q^k S^k, \end{cases} \tag{7}$$

where $\hat{\Sigma}^{k|k-1}$ and $\hat{\Sigma}^{k|k}$ denote the intermediate and the final estimations of the covaraince matrixes in the $k$-th layer respectively. In the original Kalman Filtering process, the transition matrix $\mathbf{A}^k$, the covariance matrix $R$, and the gain value $q^k$ are computed from hand-crafted formulations, but in Eqn.(7) they are all rethought as learnable parameters in a pure data-driven manner for learning efficiency.

In CNNs, the transition matrix $\mathbf{A}^k$ equals to the convolutional filter, but both the mean $\hat{\mu}^{k-1|k-1}$ and the $\hat{\Sigma}^{k-1|k-1}$ are vectors. Applying convolution to vectors is impractical. Fortunately, the Monte-Carlo Sampling Theory [30] provides a solution. Specifically, some data $y \sim$

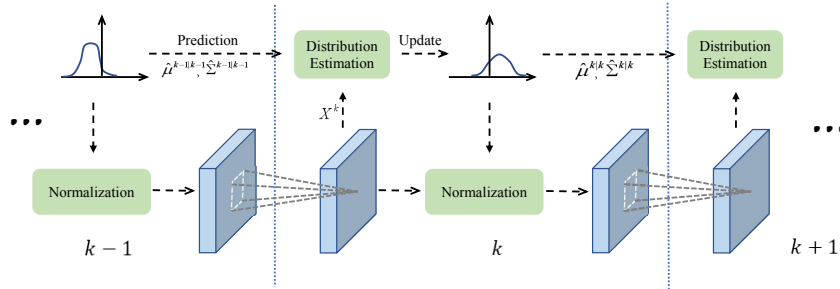

Figure 2: The estimations in the $k$-th layer ($i.e. \hat{\mu}^{k|k}$ and $\hat{\Sigma}^{k|k}$) are based on the estimations of the $(k$-1$)$-th layer ($i.e. \hat{\mu}^{k-1|k-1}$ and $\hat{\Sigma}^{k-1|k-1}$), where these estimations are updated by combining with the observed statistics of the $k$-th layer ($i.e. X^k$). This process treats the entire DNN as a whole system, different from existing works that estimated the statistics of each hidden layer independently.

$N(\hat{\mu}^{k-1|k-1}, \hat{\Sigma}^{k-1|k-1})$ is first sampled. Then, $y$ is convolved with the transition matrix $\mathbf{A}^k$ to obtain $\mathbf{A}^k y$. Finally, the intermediate estimations $\hat{\mu}^{k|k-1}$ and $\hat{\Sigma}^{k|k-1}$ are obtained by calculating the mean and the variance of $\mathbf{A}^k y$.

In training of KN, we employ $\hat{\mu}^{k|k}$ and $\hat{\Sigma}^{k|k}$ to normalize the hidden representation. Similar to BN, KN also retains the moving average statistics to appropriate the population statistics in each training iteration, and employs them during the inference.

From the above, KN has two unique characteristics that distinguish it from BN. First, it offers a better estimation of the distribution. In contrast to the existing normalization methods, the depth information is explicitly exploited in KN. For instance, the prior message of the distribution of the input image data is leveraged to improve estimation of the second layer's statistics. On the contrary, ignoring the sequential dependence of the network flow requires larger batch size. Second, KN offers a more stable estimation when learning proceeds, where the information flow from prior state to the current state becomes more stable.

Fig.2 illustrates a diagram of KN. Unlike BN where statistics are computed only within each layer independently, KN uses messages from all proceeding layers to improve the statistic estimations in the current layer.

### 3.3 Generalized Kalman Normalization

KN can also serve as an essential component. It is not specially designed for only BN, it can be combined with different BN variants. Without loss of generality, we rewrite Eqn. 1 as,

$$\bar{x}^k \leftarrow \frac{1}{m} \sum_{g \in S_i} x_g^k, \quad S^k \leftarrow \frac{1}{m} \sum_{g \in S_i} (x_g^k - \bar{x}^k)(x_g^k - \bar{x}^k)^T. \tag{8}$$

where $S_i$ is the set of pixels in which the mean/variance are computed. Specifically, in BN the set $S_i$ is defined as $S_i = \{g_C = i_C\}$ with $i_C$ as the sub-index of $i$ along the channel axis $C$. Similarly, in GN [34] $S_i$ is defined as $S_i = \{g|g_N = i_N, \frac{g_C}{C/G} = \frac{i_C}{C/G}\}$, where $G$ is a hyper-parameter and $N$ denotes the batch axis. Once obtaining $\bar{x}^k$ and $S^k$, we immediately equip them with Kalman Normalization using Eqn. 7. Different definitions of $S_i$ bring different Kalman Normalization, such as **Batch Kalman Normalization** (BKN, or KN by default) and **Group Kalman Normalization** (GKN).

### 3.4 Kalman Normalization Property

**Handling micro-batch training.** In a convolutional layer, activations of the same feature map at different locations (pixels) should be normalized in the same way. Therefore, we jointly normalize all the activations in a mini-batch over all locations (pixels) by following BN. Suppose that a layer has a mini-batch of $n$ and its feature maps have $p$ pixels, its effective mini-batch to normalization is $n \times p$ rather than only $n$.

This reveals another benefit of KN. According to Eqn.(7), the mean of the $l$-th layer can be computed as $\hat{\mu}^{l|l} = p^l \mathbf{A}^l \hat{\mu}^{l-1|l-1} + q^l \bar{x}^l$. We rewrite it as $\hat{\mu}^{l|l} = g(\hat{\mu}^{l-1|l-1}, \bar{x}^l)$. And $\hat{\mu}^{l-1|l-1}$ can be further decomposed by using the estimations of in the previous $(l-2)$ layers. Recursively, we have $\hat{\mu}^{l|l} = g(\hat{\mu}^{0|0}, \bar{x}^1, \bar{x}^2, ..., \bar{x}^l)$, where $\hat{\mu}^{0|0}$ denotes the mean of the whole dataset. This implies that in order to compute the statistics of the $l$-th layer, we achieve it by implicitly using the feature maps of all layers below, *i.e.*the effective mini-batch becomes $n \times (p^1 + p^2 + ... + p^l)$ rather than only $n \times p$, where $p^l$ denotes the number of the pixels in the $l$-th layer's feature map. In this way we enlarge the effective batch size to handle the micro-batch training.

**Micro-batch training vs data parallelism vs model parallelism.** Usually, data parallelism with a large batch size is still a micro-batch training scenario, since statistical estimation in BN need to be performed in each single GPU separately. This is different from averaging gradients in SGD: synchronizing gradients in SGD is cheap, but synchronizing the statistic in BN is expensive. In the former, all GPUs only need to wait once after each iteration, while in the latter, all GPUs need to wait at each BN layer. Given a network with 100 BN layers, there will be 100× more communication cost, making statistics synchronization in BN impractical. Unless otherwise specified, the "batch size" in the paper refers to mini-batch in a single GPU. For example, typically batch size of 32 samples/GPU is used to train a ImageNet model. Normalizations are accomplished within each GPU, and the gradients are aggregated over 8 GPUs to update the network parameters.

Similarly, model parallelism is also impractical for BN. To enable large-batch training, there are two ways to parallelize the model. i) The network is split by layer into GPUs. Without losing accuracy, we should forward-pass the data GPU by GPU, then back-propagate the errors GPU by GPU. This is inefficient due to the waiting & communicating time . ii) The network is split by channel. By blocking the information exchange between channels, the accuracy drops. The compromise between efficiency and accuracy makes model parallelism impractical for BN.

There are many typical memory-consuming scenarios that benefits from micro-batches training, such as training large-scale wide and deep networks and semantic image segmentation. Video-related problems (e.g. video detection) and object detection frameworks (e.g. Faster R-CNN [23] and Mask R-CNN [9]) are more eager for micro-batch, where batch size is typically small (<2) in each GPU.

**Comparison with shortcuts in ResNet.** Although shortcut connection also incorporates information from previous layers, KN has two unique characteristics that distinguish it from shortcut connection. 1) KN provides better statistic estimation. In shortcut connection, the informations of previous layer and current layer are simply summed up. No distribution estimation is performed. 2) In theory KN can be applied to shortcut connection, because we have received the entire feature map, then we can easily obtain the mean/variance from the feature map.

# 4 Experiments

## 4.1 ImageNet Classification

We first evaluate KN on ImageNet 2012 classification dataset [24] which consists of $1,000$ categories. The models are trained on the 1.28M training images and evaluated on the 50k validation images. We examine top-1 accuracy. Our baseline models are three representative networks, including Inceptionv2 [27], ResNet50, and ResNet101 [8]. In the original models, BN is stacked after convolution and before the ReLU activation [19]. KN is applied by simply replacing BN. We also compare with the recently proposed BRN [12] and GN [34], which can be applied in a similar manner.

### 4.1.1 Training with Typical Batch (Batch Size = 32)

Table 1 compares the top-1 validation accuracies.When reaching 73.1% accuracy for Inceptionv2, KN requires 41.2% times fewer steps than BN (100k vs 170k steps). In particular, Inceptionv2+KN achieves an advanced accuracy of 74.0% when training converged, outperform-

|  | Inceptionv2 | Iters@73.1% | ResNet101 | ResNet50 |
|---|---|---|---|---|
| BN | 73.1 | 170k | 77.4 | 76.4 |
| GN | – | – | – | $75.9^{\downarrow 0.5}$ |
| **KN** | $\mathbf{74.0}^{\uparrow 0.9}$ | **100k** | $\mathbf{78.3}^{\uparrow 0.9}$ | $\mathbf{76.8}^{\uparrow 0.4}$ |

Table 1: ImageNet *val* top-1 accuracy, $batchsize$=32.

ing the original network [13] by 1.0% . This improvement is attributed to two reasons. First, by leveraging the messages from the previous layers, estimation of the statistics is more stable in KN, making training converged faster, especially in the early stage. Second, this procedure also reduces the internal covariance shift, leading to discriminative representation learning and hence improving classification accuracy. Similar phenomenon can also be observed in ResNets. For example, KN achieves 78.3% top-1 accuracy while BN achieves only 77.4% in ResNet101.

A nonnegligible **founding** is that when compared to BN in typical-batch training, KN keeps competitive advantage (76.8% vs 76.4% in ResNet50) while GN is at a disadvantage (75.9% vs 76.4%). This may be attributed to optimization efficiency of BN, upon which KN (*i.e.*BKN) is built.

|  | Inceptionv2 | ResNet101 | ResNet50 |
|---|---|---|---|
| BN | 11.29M | 44.55M | 25.56M |
| GN | 11.29M | 44.55M | 25.56M |
| **KN** | **11.30M** | **44.60M** | **25.58M** |

Table 2: Parameter comparison.

**Extra Parameters.** In fact, KN introduces only $0.1\% \times$ extra parameters, which is negligible. The extra parameters include the gain value $q$ that is a scalar, as well as the covariance matrix $R$, which is a diagonal matrix (the same as number of channels). The parameters of KN exclude the transition matrix $\mathbf{A}$, because $\mathbf{A}$ is a state transition matrix that is shared with the convolutional filter in CNNs. An comparison of parameter numbers is shown in Table 2.

**Computation Complexity.** Table 3 reports the computation time of Inceptionv2 with KN compared to that with BN, in terms of the number of samples processed per second. For a fair comparison, both methods are trained in the same computing machine with four Titan-X GPUs. We observe that BN and KN have similar computational costs. The speed of BN is 325.74 examples/sec, which is $1.015\times$ of the speed of KN.

|  | BN | KN |
|---|---|---|
| Speed (examples/sec) | 325.74 | **320.94** |

Table 3: Computational complexity.

### 4.1.2 Training with Micro Batch (Bacth Size = 4 & 1)

Next we evaluate KN when batch size is small by using different settings, *e.g.*batch size of 1 and 4.

**Batch Size of 4.** We employ the baseline of typical batch size (*i.e.*32) for comparison. Table 4 reports the results, from which we have three major observations. First, we obtain an improvement by replacing BN with KN. For example, in ResNet50, KN achieves 76.1% top-1 accuracy, outperforming BN and BRN by a large margin (3.4% and 3.4%). Beside, KN is slightly better than GN (0.3%). This comparison verify the effectiveness of KN in micro-batch training.

|  | BN | BRN | GN | KN |
|---|---|---|---|---|
| **Option A:** using moving mean/var | 72.7 | 73,7 | – | **76.1** |
| **Option B:** using batch (online) mean/var | 75.0 | – | 75.8 | **76.1** |

Table 4: ImageNet ResNet50 val, $batchsize$=4.

Second, we also note that under such setting the validation accuracy of all normalization methods are lower than the baseline that normalized over batch size of 32 (76.8 vs 76.1 for KN), and training converges slowly. However, BN is significantly worse compared to the baseline. This indicates that the micro-batch training problem is better addressed by using KN than BN.

Third, **interestingly** we find that there is a gain between using different kinds statistic estimation. In Table 4, we compare two options including : (A) the population statistics (moving mean/variance) are used to normalize the layer's inputs during inference, and (B) batch sample (online) statistics are used for normalization during inference. Using online statistics weakens the superiority of GN over BN. This drives us to **re-think** the mechanism of 1-example-batch training (*e.g.*GN).

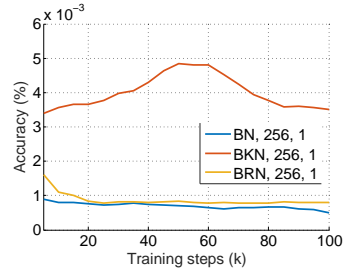

Figure 3: **Option A:** ImageNet InceptionV2 val performance using moving mean/variance.

**Batch Size of 1.** We continue to use the above two options. In return, we have two observations from in Fig. 3 and Table 5. First, in both options KN are significantly better than competitors. For example, using online statistics (B) KN obtains a 2.11% and 2.75% increase compared to BN and BRN, respectively.

Second, in comparison, using online statistics (B) is significantly better than using population statistics (A) . For example, BKN obtains a top-1 accuracy of 47.99% using online statistics (B), while 0.4% using moving means and variances. Note that this gain is solely due to the usage of different statistics. We attribute this to two reasons. 1) All approaches fail to estimate the population statistics for 1-example-batch training. As is discussed in Section 1,

|  | BN | BRN | KN |
|---|---|---|---|
| Acc @120k iters | 45.88% | 45.24% | **47.99%** |

Table 5: **Option B:** ImageNet InceptionV2 val performance using online mean/variance at 120k steps, which is not converged.

the networks are trained using batch sample statistics, while are tested based on population statistics appropriated by moving averages. In 1-example-batch training, the information communication never happens between any two examples. Therefore the moving averages are difficult to represent the population statistics. One possible solution is to also use the moving averages to normalize the layer inputs during training, but turns out to be infeasible in [13]. 2) We indeed do not need any population statistic in the case of 1-example-batch training because it ensures that the activations computed in the forward pass of training step depend only on a single example, free from the influence of population statistics. Even in Table 5 KN has a better performance than competitors, improving 2.11% and 2.75% compared to BN and BRN respectively. These results verify the effectiveness of KN.

## 4.2 COCO 2017 Object Detection and Segmentation

To investigate the application of micro-batch training, we use CO-CO 2017 detection & segmentation benchmark [6]. We evaluate fine-tuning the models trained on ImageNet [24] for transferring to detection and segmentation. These computer vision tasks in general benefit from higher-resolution input, so the batch size tends to be small in common practice (1 or 2 images/GPU). As a result, BN degrades into a linear layer $y = \frac{\gamma}{\sigma}(x - \mu) + \beta$ where $\mu$ and $\beta$ are pre-computed from pre-trained model and frozen, *e.g.*Mask RCNN [9]. We denote this as BN*, which in fact performs no normalization during finetuning. Another substitute is to use the standard BN, but it turns out to be impractical in [34] because of inaccurate statistic estimation. Therefore we ignore the standard BN.

| backbone | $AP^{bbox}$ | $AP^{mask}$ |
|---|---|---|
| BN* | 36.7 | 32.1 |
| GN | 37.7 | 32.5 |
| **KN** | **37.8** | **33.1** |

Table 6: Detection and segmentation ablation results using Mask RCNN.

We experiment on the Mask RCNN baselines [9] using a ResNet50 conv$_4$ backbone. We replace BN* with KN during finetuning. The models are trained in the COCO train2017 set and evaluated in the COCO val2017 set. To accelerate the training, we use the standard fast training setting following the COCO model zoo. Specifically, the resolution is set as (800, 1333); and we sample 256 boxes for each image. We use the schedule of 280k training steps. We report the standard COCO metrics of Average Precision (AP) for bounding box detection ($AP^{bbox}$) and instance segmentation ($AP^{mask}$).

Table 6 shows the comparison of KN *vs* BN* *vs* GN. KN improves over BN* by 1.1% box AP and 1.0% mask AP. This may be contributed to the fact that BN* creates inconsistency between pre-training and fine-tuning (frozen). We also found GN is 0.6% mask AP worse than KN. Although GN is also suitable for micro-batch training, its representational power is weaker than KN.

## 4.3 Analysis on CIFAR10, CIFAR100, and SVHN

We conducted more studies on the CIFAR-10 and CIFAR-100 dataset [15], both of which consist of 50k training images and 10k testing images in 100 classes and 10 classes, respectively. We also conduct experiments on SVHN dataset [20], which is a real-world digit image dataset containing over 600,000 labeled data of 10 categories.

### 4.3.1 Generalized Kalman Normalization Studies

As is pointed out in Sect. 3.3, there are various Kalman Normalizations, *e.g.*BKN and GKN. Next we investigate the gain of Kalman Normalization mechanism compared with the bare BN and GN on CIFAR10. We use the standard ResNet for CIFAR10 following [8] with the setting of $n = 5$. We conduct the experiments in the context of micro-batch training, *i.e.*we use batch size of only 2. The results are reported in Figure 4, where we have three major observations. First, both BN and GN benefit from Kalman Normalization mechanism. For example, BKN has a gain of 1.5% compared with BN, verifying the effectiveness of BKN (*i.e.*KN). Second, the gain of 'BKN - BN' is larger than 'GKN - GN' (1.5% vs 0.4%). This may

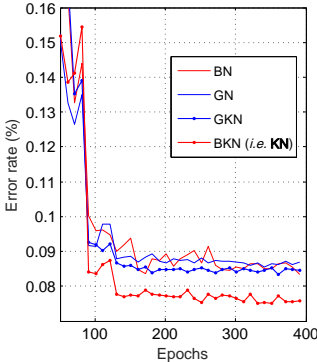

Figure 4: Comparison among BN, BKN, GN, and GKN on CIFAR-10 *val* set, ResNet(n=5)

be attributed to optimization efficiency of BN. Third, Although GN has gains over BN on ImageNet in micrio-batch training, it has no gain on CIFAR10.

### 4.3.2 Other Ablation Studies

In this section our focus is on the behaviors of extremely small batch size, but not on pushing the state-of-the-art results, so we use simple architecture summarized in the following table, where a fully connected layer with 1,000 output channels is omitted.

| type | conv | inception | inception | inception | avg pool |
|---|---|---|---|---|---|
| spatial size | $16 \times 16$ | $16 \times 16$ | $16 \times 16$ | $16 \times 16$ | $1 \times 1$ |
| filters | 32 | 256 | 480 | 512 | 512 |
| $1\times1$ | | 64 | 128 | 192 | |
| $1\times1/3\times3$ | | 96, 128 | 128, 192 | 96, 208 | |
| $1\times1/5\times5$ | | 16, 32 | 32, 96 | 16, 48 | |
| pool/$1 \times 1$ | | 32 | 64 | 64 | |

**Evidence of more accurate statistic estimations.** To show that KN indeed provides a more accurate statistic estimation than BN, we present two evidences as follows. First, direct evidence. When the training stage finished, we exhaustively forward-propagated all the samples in CIFAR-10 to obtain their moving statistics and batch sample statistics. The gaps between batch sample variance and the moving variance are visualized in Fig. 5 (a) and (b) for BN and KN, respectively. In Fig. 5 the horizontal axis represents values of different batches, while vertical axis represents neurons of different channels. We can observe

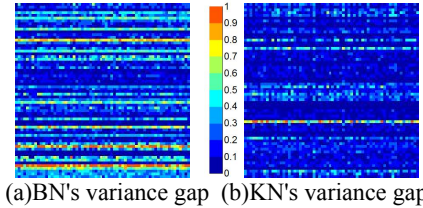

(a)BN's variance gap   (b)KN's variance gap

Figure 5: Visualization of variance gap between batch sample variance and moving variance for BN and KN, respectively.

values in Fig. 5 (b) are smaller than Fig. 5 (a), indicating that KN provides a more accurate statistic estimation, which is consistent with Table 4. This reflects the superiority of KN over BN. The improvement is attributed to two reasons. First, KN enlarges the effective batch size to handle the micro-batch training by implicitly using the feature maps of all preceding layers (see Sec.3.3). Therefore it provides a more accurate statistic estimation (*i.e.*smaller gap between population statistic and sample statistic). Second, BN treats each hidden layer as an isolated system, the gap between the population variance and the batch sample variance amplifies as the network becomes deeper. Differently, KN treats all the layers in a network as a whole system, and estimates the variance of a certain layer guided by the distributions of its preceding layer. The merits of Kalman Filtering help eliminate gaps.

Second, indirect evidence. During inference, there are two ways to calculate the classification accuracy, i.e. using the moving mean/variance or batch mean/variance. Experimental results in Table 7 show that in KN, using batch mean/variance achieves the same

|  | using online mean/var | using moving mean/var |
|---|---|---|
| BN ($bs = 2$) | 90.0 | 89.4 |
| BN ($bs = 128$) | 90.0 | 92.1 |
| **KN** ($bs = 2$) | **90.9** | **90.9** |

Table 7: CIFAR-10 *val* set, $bs = batchsize$, Inception.

accuracy as using moving mean/variance. While in BN there's a gap between using batch variance and moving variance. This again proves that KN does provide more accurate estimations.

**Comparison with BN variants.** We compare KN with more BN variants (*e.g.*Batch Renorm(BRN) [12], Weigth Norm (WN) [25], Layer Norm (LN) [2] and Group Norm (GN) [34] ) on CIFAR-10, CIFAR-100 and SVHN dataset. We have three major findings in Table 8. First, KN beats BN and its variants by large margin on these dataset in micro-batch training. For example, on CIFAR100 KN has a gain of 3.5%, 1.58%, 5.3%, 20.3% and 1.4% when compared with BN, BRN, WN, LN, and GN, respectively. Second, we can observe that the performance of the micro-batch training (91.0%, $batchsize = 2$) is very encouraging compared to that of the typical size (92.1%, $batchsize = 128$). Third, different from GN that is inferior to BN under the context of typically large-batch training, KN keeps superiority over the competitors. These comparisons verify the effectiveness of KN again.

| Inception | | | |
|---|---|---|---|
|  | CIFAR10 | CIFAR100 | SVHN |
| BN ($bs = 2$) | 89.4 | 63.8 | 98.06 |
| BRN [12]($bs = 2$) | 90.38 | 65.72 | 98.04 |
| WN [25] ($bs = 2$) | 87.83 | 62 | 97.92 |
| LN [2] ($bs = 2$) | 77.7 | 47.02 | 97.98 |
| BN ($bs = 128$) | 92.1 | 70.5 | 98.08 |
| **KN** ($bs = 2$) | **90.9** | **67.3** | **98.16** |
| ResNet32 | | | |
|  | CIFAR10 | | |
| GN ($bs = 2$) | 91.3 | | |
| BN ($bs = 2$) | 91.2 | | |
| **KN** ($bs = 2$) | **92.7** | | |
| ResNet110 | | | |
|  | CIFAR10 | | |
| GN ($bs = 128$) | 92.6 | | |
| BN ($bs = 128$) | 93.8 | | |
| **KN** ($bs = 128$) | 94.3 | | |

Table 8: Comparison with BN variants on CIFAR10, CIFAR100 and SVHN, $bs = batchsize$

## 5   Conclusion

This paper presented a novel normalization method, called Kalman Normalization(KN), to normalize the hidden representation of a deep neural network. Unlike previous methods that normalized each hidden layer independently, KN treats the entire network as a whole. KN can be naturally generalized to other existing normalization methods to obtain gains. Extensive experiments suggest that KN is capable of strengthening several state-of-the-art neural networks by improving their training stability and convergence speed. More importantly, KN can handle the training with mini-batches of very small sizes.

**Acknowledgments**

This work was supported in part by the National Key Research and Development Program of China under Grant No. 2018YFC0830103, in part by National High Level Talents Special Support Plan (Ten Thousand Talents Program), and in part by National Natural Science Foundation of China (NSFC) under Grant No. 61622214, and 61503366.

## Footnotes

*Corresponding author: Liang Lin.

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
