[Reviews · NeurIPS 2018]

Reviewer 1



This paper addresses the batch normalization problem with small batch size. Traditional batch normalization relies on the estimation of the batch statistics (mean and variance) over each batch. Batch normalization requires relatively large batch size in order to obtain relatively reliable estimates of these statistics. However, due to memory constraint, some higher-level tasks (e.g., in computer vision) could not use large batchsize. Therefore, developing normalization method with small batch size is important for improving the performance of these systems. Experient results show that the proposed KN outperform previous batch normalization approaches both in regular batch sizes and small batch sizes. The problem formulation and the proposed algorithm have a gap from the canonical Kalman filter and it is not discussed clearly in the paper. It is a bit confusing about the formulation of the Kalman filter. What is the state x^k in (2)? Is it the activation vector at the k-th hidden layer or the groundtruth mean at the hidden layer? I think a correct formulation of Kalman filter for this problem is to model the state as the true mean value if the observation is modeled to be the (noisy) observation of this mean and u^k is the disturbance due to inaccurate modeling with a linear dynamic system. Otherwise, if the state is modeled to be the realization of the hidden activation vector at the k-th layer, then \hat{\mu}_{k|k}, which in Kalman filter context is the conditional mean of the state given all the observations up to k, is an minimum-mean-square-error (MMSE) estimate of the realization of the activation vector. In batch normalization problem, the desired quantity to estimate should be the mean not the realization of the activation vector. So the current formulation in (2) is not correct. Furthermore, the proposed algorithm is quite different from standard Kalman filter, where the gain matrix (instead of a gain scalar q^k) is computed from a Riccati equation (a nonlinear recursion over the covariance matrix). None of these are sufficiently discussed.

Reviewer 2



This paper introduced a new normalization method for improving and accelerating DNN training. The proposed method, Kalman Normalization (KN) estimates the statistic of a layer with the information of all its preceding layers. This idea is vary similar to to shortcut link in the ResNet, however, KN provides better statistic estimation. The experiments presented solid results and conclusions that show KN not only improves training accuracy, but also help convergence as well. Strength of this paper: 1. The proposed normalization method is new, although it mimic the process of Kalman filter and has similar idea as ResNet. This approach shows a new angel and as experiment showed, very effective. 2. The experiments are well designed. The author not only compared with the state-of-the-art results , but also showed the evidence of more accurate statistic estimations that KN can provide. Some mirror suggestions: 1. In experiment section 4.1.1, it would be interesting to report how many step did KN use to get the best performance (74%) 2.Typo: Table 5 AP^{maks} ->A^{mask} 3. In Figure 5, the vertical axis represents neurons if different channels, and we can see that there are some kind of channel patterns: can you explain why some channels have very bad gap in (a) but seems to be a lot better with KN? Overall, I believe this is a paper with high quality and the proposed normalization method will help different applications in DNN area.

Reviewer 3



This paper presents a normalization method, called Kalman Normalization (KN), which is a variant of Batch Normalization (BN). It aims to alleviate the issue of BN in case like the micro batch size, which is hard to estimate true means and variances. The solution is to utilize the statistics of preceding layers. The empirical results seem good(e.g., Table 1). My main concern is that KN introduces many learnable parameters. For instance, in Eq. 6, the transition matrix A, the covariance matrix R and the gain value q are required to learn for each layer during the training. Thus, in the case of large neural networks as line 202, it is hard to fit the model in GPUs. I’d also like to see the parameters comparison in Table 1. Practically, for large neural networks, model parallelism is an option as well. In the extreme case, such as batch size is 1, I am wondering if it is necessary to use BN/KN. What’s the performance without any normalization?